# Artificial Insemination as a Possible Convenient Tool to Acquire Genome-Edited Mice via In Vivo Fertilization with Engineered Sperm

**DOI:** 10.3390/biotech13040045

**Published:** 2024-11-11

**Authors:** Masahiro Sato, Emi Inada, Issei Saitoh, Kazunori Morohoshi, Shingo Nakamura

**Affiliations:** 1Department of Genome Medicine, National Center for Child Health and Development, Tokyo 157-8535, Japan; 2Department of Pediatric Dentistry, Graduate School of Medical and Dental Sciences, Kagoshima University, Kagoshima 890-8544, Japan; 3Department of Pediatric Dentistry, Asahi University School of Dentistry, Gifu 501-0296, Japan; 4Division of Biomedical Engineering, National Defense Medical College Research Institute, Saitama 359-8513, Japan

**Keywords:** artificial insemination, in vitro fertilization, genome editing, engineered sperm, sperm-mediated gene transfer, genome-edited animals

## Abstract

Advances in genome editing technology have made it possible to create genome-edited (GE) animals, which are useful for identifying isolated genes and producing models of human diseases within a short period of time. The production of GE animals mainly relies on the gene manipulation of pre-implantation embryos, such as fertilized eggs and two-cell embryos, which can usually be achieved by the microinjection of nucleic acids, electroporation in the presence of nucleic acids, or infection with viral vectors, such as adeno-associated viruses. In contrast, GE animals can theoretically be generated by fertilizing ovulated oocytes with GE sperm. However, there are only a few reports showing the successful production of GE animals using GE sperm. Artificial insemination (AI) is an assisted reproduction technology based on the introduction of isolated sperm into the female reproductive tract, such as the uterine horn or oviductal lumen, for the in vivo fertilization of ovulated oocytes. This approach is simpler than the in vitro fertilization-based production of offspring, as the latter always requires an egg transfer to recipient females, which is labor-intensive and time-consuming. In this review, we summarize the various methods for AI reported so far, the history of sperm-mediated gene transfer, a technology to produce genetically engineered animals through in vivo fertilization with sperm carrying exogenous DNA, and finally describe the possibility of AI-mediated creation of GE animals using GE sperm.

## 1. Introduction

From the mid-2000s to the early 2010s, technologies targeting mutagenesis at a target locus using nuclease-based genome-editing tools, such as meganucleases [1], zinc-finger nucleases (ZFNs) [2,3,4], and transcription activator-like effector nucleases (TALENs) [5,6,7], were developed. In late 2013, the creation of genome-edited (GE) animals using genome-editing technology was substantially advanced using clustered, regularly interspaced short palindromic repeats (CRISPR)/CRISPR-associated (Cas)9 (CRISPR/Cas9), which is more convenient and simpler than other methods, such as ZFNs and TALENs [8,9].

The CRISPR/Cas9 system employs only two factors, a guide RNA (gRNA) and a Cas9 endonuclease. The former can bind to a specific chromosomal DNA site, and the latter can cleave double-stranded (ds) DNA three bases upstream of the protospacer adjacent motif (PAM) under the guidance of gRNA [8,9]. When dsDNA is cleaved by the Cas9 protein, double-strand breaks (DSBs) occur at the target site. In the absence of a donor DNA sequence, the resulting DSBs are usually repaired via a nonhomologous end-joining (NHEJ)-based process, which frequently leads to nucleotide insertions or deletions (indels) at the cleaved site. These indels often generate stop codons causing premature termination, leading to the failure of protein expression or nonsense mutations. In the former case, the emerging abnormal mRNA is often destroyed through nonsense-mediated mRNA decay, a translation-dependent surveillance mechanism in eukaryotes [10]. Consequently, animals with indels at a target locus exhibit a knockout (KO) or loss-of-function phenotype. In contrast, when DSBs generated by Cas9-mediated cleavage are repaired in the presence of a donor DNA sequence, homology-directed repair (HDR) occurs at the cleavage site. However, the efficiency of HDR-mediated KI of the donor sequence is typically lower than that of NHEJ-mediated genome editing. This event is called HDR-mediated knock-in of the donor sequence. Notably, HDR preferentially occurs in dividing cells, whereas NHEJ occurs in both dividing and non-dividing cells [11]. Although NHEJ inhibitors have been employed to elevate HDR efficiency through the suppression of the NHEJ pathway in cultured cells [12,13], to the best of our knowledge, there is no standardized method that can control the efficiency of HDR in early mammalian embryos [14].

Base editing (or editor) (BE) and primer editing (PE) systems, recently developed and building upon CRISPR/Cas9 to introduce targeted changes in the DNA, have now been developed [15,16]. BE allows single-base conversions without the need for a DSB. Two classes of DNA base-editors, cytosine BEs (CBEs) and adenine BEs (ABEs), have been described. Both systems of base editing have been applied successfully to zygotes, resulting in the generation of GE animal models. For instance, Kim et al. [17] first demonstrated the successful creation of animal models using the MI of BE3 mRNA and sgRNA in mouse embryos. In this system, KO mutations are achieved by C-to-T editing within a coding exon to create a premature stop codon. Consequently, 55% of the founder mice had mutations in the Duchenne muscular dystrophy (*Dmd*) gene. Despite C-to-T conversion being the most common mutagenic outcome, alleles that frequently show additional mutations, including deletions at the target site, have been generated [17]. PE was originally developed as a simple ‘search and replace’ editing tool, while it does not require the exogenous donor DNA and the DSBs [18]. Compared with BE technology, the editing scope of PE has been widely expanded. The PE system consists of a fusion protein of Cas9 nickase (nCas9; H840A) and Moloney murine leukemia virus reverse transcriptase along with a prime editing guide RNA (pegRNA) [18]. Currently, PE systems are used in mice [19]. Liu et al. [19] showed that the frequency of base conversion mediated by PE3 in *the homeobox D13* (*Hoxd13*) locus in mouse embryos ranged from 1.1 to 18.5%, which appears to be too low for application in animals. Qian et al. [20] employed an updated ePE3max system [comprising a PEmax protein, *engineered* pegRNA (epegRNA) with structured RNA motifs (evopreQ1), and nicking sgRNA] to induce gene editing in mouse and rabbit embryos. They successfully generated a *Hoxd13* (c. 671 G  >  T) mutation in mice and a *tyrosinase (Tyr*) (c. 572 del) mutation in rabbits using the ePE3max system with efficiencies of 43% and 70–100%, respectively.

The creation of GE mammalian zygotes was first performed using pronuclear or cytoplasmic MI of genome editing reagents and zygotes isolated from pregnant female rodents or those obtained through in vitro fertilization (IVF) [21,22,23,24,25,26,27]. However, this approach requires an expensive micromanipulator for MI and a skilled person to manipulate the embryos under a microscope; therefore, it takes a long time to complete the process. It typically takes over 2 h to treat approximately 100 zygotes. In 2014, Kaneko et al. [28] developed a convenient and efficient approach using in vitro electroporation (EP) to create GE rats. In contrast to the MI-mediated production of GE animals, in vitro EP requires an expensive electroporator; however, this method does not require skilled staff, unlike MI, and many zygotes (>30) can be simultaneously GE. This approach has been effective in mouse zygotes [29,30,31,32,33,34,35], rat zygotes [36,37,38,39], and porcine embryos [40,41,42,43,44]. Notably, both systems require the ex vivo handling of embryos (i.e., isolation of zygotes from a pregnant mother to perform genome editing in vitro, culture of treated embryos for a short period, and embryo transfer (ET) of GE embryos to the reproductive tract of a pseudopregnant recipient female). In particular, ET requires a highly skilled staff and recipient females whose estrous cycles have been synchronized after mating with vasectomized males. In 2015, to bypass the above ex vivo handling of embryos, a novel technique enabling genome editing in early mouse embryos via oviductal nucleic acid delivery (GONAD), was developed [45]. In this system, a small volume (1–1.5 μL) of a solution containing genome-editing reagents (gRNA and *Cas9* mRNA) and trypan blue (a visible marker for monitoring the injection process) was injected via the oviductal wall into the ampulla of the oviduct of a pregnant female (corresponding to the two-cell stage) using a mouthpiece-controlled glass micropipette. Immediately after injection, the entire oviduct was held using tweezer-type electrodes and electroporated using a square-pulse generator (electroporator). Under appropriate electrical conditions, the genome-editing components present outside the embryo (floating in the oviductal lumen) are transferred via the zona pellucida (ZP) to the inside of the embryo. In 2018, Ohtsuka et al. [46] improved GONAD and re-named it “improved GONAD (*i*-GONAD)”. This revised system was aimed at targeting late zygotes to avoid possible mosaicism and employed ribonucleoprotein (RNP) (comprising Cas9 protein and crRNA/tracrRNA) for the efficient induction of genome editing more rapidly than when using *Cas9* mRNA [46]. GONAD/*i*-GONAD does not require the ex vivo handling of embryos and is, therefore, considered more convenient than previous methods based on MI or in vitro EP. One session required only four to six pregnant females and was completed within 15 min for both oviducts. This fits the 3R principle (replacement, reduction, and refinement), an international tenet for animal experimentation intended to reduce the number of animals used.

As mentioned above, most experiments have focused on the creation of GE animals (i.e., mice and rats) using preimplantation embryos, such as fertilized eggs or two-cell embryos. However, these preimplantation embryo-targeted approaches for genome editing are often considered challenging in large animals, such as pigs and cows, because it is widely recognized that it is difficult to obtain early embryos from these animals for in vitro genome editing or to perform ET of GE embryos to the reproductive tract of a recipient female for further development. Therefore, gene delivery to the sperm appears to be simpler than targeting early embryos. Various previously established methods to introduce genes into mature sperm employ various gene delivery-enhancing reagents, such as micronanoparticles and chemicals. It is highly likely that these genetically engineered sperm can be used as vehicles for delivering GE traits to oocytes through fertilization. Therefore, these sperms can be subjected to fertilization in vitro or in vivo through IVF or artificial insemination (AI), respectively, to obtain GE pups.

## 2. IVF-Based Production of Gene-Engineered Animals

In 1989, Lavitrano et al. [47] reported a simple, convenient, and cost-effective method to create GM animals, sperm-mediated gene transfer (SMGT), where isolated epididymal sperms were incubated in the presence of naked plasmid DNA for a short period and these DNA-associated sperms were subjected to IVF with normal oocytes (Figure 1). The resulting progeny were later determined to carry exogenous DNA into their genome. Since this report, its reproducibility has been controversial [48,49,50]. Sciamanna and Spadafora [51] speculated that most SMGT-related experiments are frequently associated with extrachromosomal arrangements, low copy numbers, and a mosaic distribution of foreign DNA among organs and organisms. These characteristics contribute to the low efficiency and reproducibility of this technique for creating transgenic (Tg) animals [52,53,54]. Nevertheless, SMGT has substantial potential for producing Tg animals in a cost-effective and efficient manner. Kang et al. [55] demonstrated that sperm can naturally bind to nucleic acids, but their binding ability is inhibited after seminal fluid removal. This suggests that additional enhancements are required to boost their effectiveness (the ability of sperm to tightly bind to nucleic acids).

Numerous studies have recently proposed techniques to enhance the DNA uptake by sperm cells as modified methods of SMGT (Table 1). These include linker-based technique [56], viral vectors [57], and reagents, such as dimethyl sulfoxide (DMSO) [58], magnetic nanoparticles (MagNPs) [59,60], restriction enzyme-mediated insertion (REMI) [61], zeolitic imidazolate framework-8 (ZIF-8) [62], and methyl β-cyclodextrin (MBCD), a family of β-cyclodextrins (βCDs) [63]. Electrolyte-free medium (EFM) (which is composed of 0.33 M glucose and 3% bovine serum albumin and used for short-term storage of human spermatozoa without freezing) is also used to increase the rate of SMGT when DNA/DMSO are included in the EFM medium [64]. Notably, Shen et al. [58] first demonstrated the usefulness of DMSO for accelerating the DNA uptake by sperm. When mouse sperm were incubated in a solution containing 3% DMSO and 20 ng/µL of plasmid DNA for 10–15 min at 4 °C prior to IVF, the resulting embryos (42%, 25/60) showed bright, enhanced green fluorescent protein (EGFP)-derived fluorescence [58]. Furthermore, Kim et al. [59] demonstrated that magnetic nanoparticles can be excellent vehicles for enhancing gene delivery to sperm from various animals, such as boars. When boar sperm (10^7^) were incubated in 200 μL of solution composed of 160 ng of plasmid DNA (encoding green fluorescent protein (GFP)) and 0.5% (*v*/*v*) of MagNPs on the magnetic field for 90 min, and the magnetofected sperm were subjected to IVF with normal oocytes, the resulting fertilized eggs expressed GFP at the morula stage. PCR analysis demonstrated that all samples (morulae and blastocysts) tested contained the introduced transgene. These results imply that exogenous DNA can be transferred into fertilized oocytes using magnetofected spermatozoa. Kim et al. [59] concluded that the magnetofection technique could improve the efficiency of SMGT for livestock Tg applications. Recently, Sameni et al. [62] used ZIF-8, a porous metal-organic framework, in SMGT to enhance gene delivery to sperm cells. First, a combination of ZIF-8 (60 ng/µL) and GFP-expressing plasmid (20 ng/µL) was produced. After incubation for 15 min, the mixture was exposed to sperm for an additional 30 min incubation period within an incubator. Sameni et al. [63] confirmed the successful generation of Tg blastocysts because they exhibited *EGFP* cDNA-derived fluorescence. These findings demonstrate that forming ZIF-8/DNA complexes between ZIF-8 nanoparticles and plasmid DNA enables the uptake of exogenous DNA by mouse sperm cells.

These results show the feasibility of using SMGT to acquire Tg embryos or pups when gene delivery-enhancing reagents are employed during IVF. However, it required ET of these IVF-derived gene-engineered embryos to recipient females for further development, which is laborious and requires specialized skills. To bypass this process, it is theoretically possible to perform AI using in vitro-transfected sperm, which can be performed by injecting the sperm into the uterine horn, lumen of the oviducts, or a space near the infundibulum between the ovary and the ovarian bursa of recipient females showing oocyte ovulation (Figure 2). We previously called this approach SMGT-based AI (SMGT-AI), for the purpose of gene-engineered animal production [65].

## 3. IVF vs. AI

AI is an assisted reproduction technology (ART) based on the introduction of isolated sperm into the female reproductive tract, such as the uterine horn or oviductal lumen, for the in vivo fertilization of ovulated oocytes. IVF is also an ART based on the in vitro fertilization of oocytes and freshly isolated or cryopreserved sperm. Both techniques have been used to generate viable embryos. In the case of AI, fertilization occurs in vivo and subsequent embryonic development continues in the recipient female, whereas embryos generated from IVF must be transferred to a suitable pseudopregnant recipient for in vivo development, which is called “ET”, a time-consuming and challenging technique. IVF is always combined with ET for making viable pups; therefore, we hereinafter call IVF “IVF-ET”.

AI is simpler than IVF-ET, because sperm are directly added to the reproductive tract of the recipient female, and further development occurs in vivo. Therefore, AI can be an alternative to IVF. However, historically, AI has not been as extensively employed as IVF-ET, possibly because AI in mice has variable success rates and tends to be complicated [66].

AI can be performed either surgically or non-surgically (Figure 2). The surgical procedure usually involves the direct injection of a small volume of sperm into the oviduct (intraoviductal transfer of spermatozoa [IOTS]) [67,68,69] or the space between the ovarian bursa and infundibulum (intrabursal transfer of spermatozoa [ITS]) [69,70] to produce viable embryos (Figure 2A). It requires the opening of the ventral skin to reach the female reproductive tract and subsequent exposure of the ovary/oviduct/uterus on the skin surface of an anesthetized mouse. Using a mouth-controlled glass micropipette, a small volume (<2 μL) of sperm-containing solution is injected into the oviductal lumen (IOTS) or a space between the ovarian bursa and infundibulum (ITS) under observation using a stereo microscope (Figure 2A). Sato et al. [69] compared in vivo fertilization rates of IOTS and ITS. When 1 μL of solution containing spermatozoa (~1 × 10^5^) freshly isolated from B6C3F1 (a hybrid between C57BL/6 and C3H/HeN mice) males were injected into superovulated B6C3F1 females on E 0.4 (10:00 am; 17 h after hCG administration; E 0 is defined as the day when copulation plug is detected), normal two-cell embryos were recovered from the females at rates ranging from 14 to 23% with each method. This rate was much lower than that of the embryos obtained through natural mating (approximately 93%). Furthermore, in both cases, females given exogenous spermatozoa yielded mid-gestational fetuses with average litter sizes of 2.5 and 2.8. Sato et al. [71] ascribed this low fertilization rate to a delay in the timing of in vivo fertilization. They performed AI (ITS) by injecting fresh B6C3F1 epididymal sperm into the space between the infundibulum and the ovarian bursa 1, 7, 12, or 17 h after hCG administration. When the number of developing embryos recovered from the oviducts was checked, the ITS 7 h after hCG administration yielded the highest rate (75%) of cleaved embryos, which was comparable (*p* > 0.05) to that (91%) of embryos obtained after natural mating (control) [71]. Furthermore, ITS 7 h after hCG administration yielded normal mid-gestational fetuses with an average litter size of seven. Based on these findings, the timing of AI is considered a key factor affecting in vivo fertilization efficiency.

In contrast, non-surgical procedures in mice were developed over 50 years ago. Leckie et al. [72] described a procedure for transcervically transferring sperm with a blunt needle using an artificial penis and a vaginal tampon (as an alternative to pairing with a vasectomized male to induce a pseudopregnant state) to increase the efficiency of embryo implantation. Notably, a non-surgical AI technique introducing sperm into the vagina without anesthesia or analgesia was recently developed by Stone et al. [66] who used females in estrus or superovulated female mice for developing a simple user-friendly AI. They also used the non-surgical embryo and sperm transfer (mNSET) device for mice, originally designed for the non-surgical transfer of embryos (blastocysts) into the uterine horns of pseudopregnant female mice [73], for sperm transfer (Figure 2B). The device consists of a flexible Teflon catheter that can pass through the cervix and deposit embryos or sperm into the uterine horn [74]. In this case, AI using the mNSET can be used as an alternative to the previous non-surgical procedure based on the use of some items, such as blunt needle, glass speculum, artificial penis, and vaginal tampon [66].

**Figure 2 biotech-13-00045-f002:**
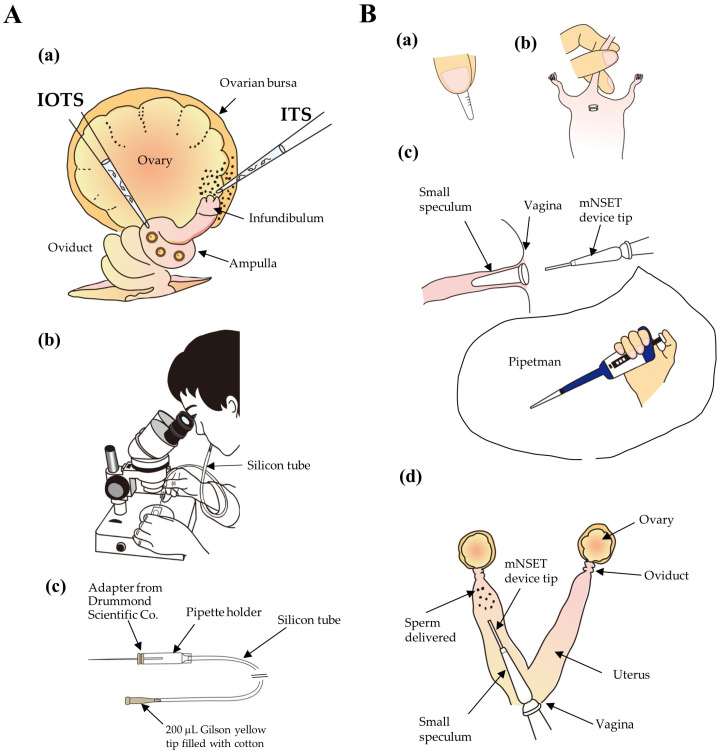
Various methods of artificial insemination (AI). (**A**) Schematic representation of surgical AI based on intraoviductal transfer of sperm, IOTS, and intrabursal transfer of sperm, ITS (**a**). Both techniques were performed under a dissecting microscope (**b**), using a mouth-controlled micropipette (**c**). This illustration is based on studies by Sato et al. [69] and Sato and Kimura [70]. (**B**) Schematic representation of non-surgical AI using the non-surgical embryo and sperm transfer (mNSET) device in unanesthetized females. First, the small speculum tip (**a**) was inserted into the vagina of the female, as shown in (**b**). Second, the tip containing 40 μL of sperm was inserted into the small speculum using Pipetman (**c**) for releasing sperm within a uterine horn (**d**). This illustration is based on Stone et al. [66] and Stone [72].

Although non-surgical AI is recognized as a rapid, non-stressful sperm transfer without the need for anesthesia or analgesia [66], it requires a large volume of sperm (>40 μL), compared to surgical AI in which only a small volume of sperm (<2 μL) is usually required [69,70,71]. Furthermore, individuals who are comfortable performing ET might find the surgical AI-based approach much easier than the non-surgical AI-based approach [75].

Several factors must be considered to successfully perform AI. First, the timing of hormone administration relative to sperm delivery is important. Notably, in the natural estrous cycle of mice, ovulation and fertilization are thought to occur around midnight during the dark cycle [76]. Based on this assumption, Sato et al. [71] proposed that the time of surgical AI should preferably be 7–12 h after administration of hCG (which corresponds to the stage of ovulation), as oocytes generated after hormone-induced ovulation are most efficiently fertilized. Therefore, successful AI may be recommended at 12 to 5 AM when PMSG-hCG is administered at 5 PM. The timing of ovulation and fertilization is inconvenient for the routine use of AI [66]. To avoid this, Sato et al. [71] suggested a shift in the hormone administration timing (Figure 3A), for example, the PMSG-hCG administration at 9 AM. After hCG administration, AI was performed between 4 and 6 PM. In this case, there was no need to change the 12 h light/12 h dark schedule (7 AM to 7 PM). Second, hormone administration can frequently be used to induce timed estrus and ovulation in oocytes for AI. However, superovulation has adverse effects on embryonic and fetal development [77,78,79]. Consistent with this, we found that no viable pups were naturally delivered when 5 IU of PMSG-5 IU of hCG was administered to B6C3F1 females, which were subsequently mated with B6C3F1 males [80]. To overcome this problem, low concentrations of PMSG have been used (2 to 0.2 IU) to induce the natural delivery of pups. This concentration has proven beneficial [80] as the hCG concentration (5 IU) remained unchanged. Kobayashi et al. [81] demonstrated that 44% of C57BL/6 females that had been administered 2 IU of PMSG and 5 IU of hCG 48 h apart, successfully delivered viable pups with an average littermate of six, which was comparable to the rate obtained by hormone-uninjected females. Coherent treatment with PMSG and hCG to induce superovulation in mice has been widely used. However, an alternative method to PMSG has recently been demonstrated. Hasegawa et al. [82] aimed to synchronize the estrous cycle in C57BL/6 female mice by injecting anti-inhibin serum (AIS) instead of PMSG. The mean number of ovulated oocytes almost doubled (41 vs. 21 per mouse). Estrous cycle synchronization, followed by AIS-hCG treatment, was also effective in other inbred (BALB/cA), outbred (ICR), and hybrid (B6D2F1) strains. Later, the same group [83] employed anti-inhibin monoclonal antibodies (AIMAs) instead of PMSG and showed that when C57BL/6 female mice were treated with AIMA and mated, the number of healthy offspring per mouse increased by 1.4-fold (11.9 vs. 8.6 in controls). In contrast, treatment with PMSG/hCG or anti-inhibin serum resulted in fewer offspring than in the non-treated controls.

Notably, the induction of pseudopregnancy in recipients is also an important factor in determining successful AI-based production of pups. In some cases, immediately after hCG administration, females were mated with vasectomized males to induce pseudopregnancy [67,68,71]. In contrast, Stone et al. [66] obtained the highest pregnancy rate through induction of pseudopregnancy by mating females with vasectomized males after AI (Figure 3B). For example, 1 IU of PMSG was administered at 5:30 PM three days prior to sperm transfer, and 1 IU of hCG was administered at 5:00 PM one day prior to sperm transfer to ovulate AI recipients. The next morning, sperm was collected at 8:00 AM, capacitated for 1 h, and 40 µL of sperm suspension was transcervically delivered at 9:00 AM to AI recipients. The recipients were immediately paired overnight with vasectomized males. Full-term pregnancy rates with fresh sperm have reached 50%. Litter size averaged 5.0 pups with all pups surviving until weaning.

The AI approach for obtaining viable pups has several advantages over the IVF-ET-based approach. For example, multiple females can be inseminated with sperm from a single male to rescue the line (e.g., Tg or mutant mice), expand the line quickly, or generate relatively synchronous embryos [75]. According to Mayer et al. [84], certain genetically engineered lines are difficult to breed and archive. AI is a useful tool for avoiding the possible extinction of a Tg allele in Tg or mutant mice with fertility problems as an alternative to IVF [66]. Additionally, sperm can be shipped globally under storage at either cold [85,86] or room temperatures [87,88,89], thereby reducing the need to ship live animals. AI can provide a means to rapidly recover a germ-free strain after shipping sperm to a *sterile environment*. Notably, Nakao et al. [90] recently employed AI in an attempt to increase the in vivo fertilization rate of ultrasuperovulated oocytes which were obtained by their improved superovulation technique, ultrasuperovulation; the administration of inhibin antiserum and equine chorionic gonadotropin [IASe]. The IASe produced 100 oocytes from a single female C57BL/6 mouse, but resulted in only approximately 20 fertilized oocytes via mating. Nakao et al. [90] speculated that this low fertilization rate could be ascribed to a sperm shortage in the ampulla. By controlling the synchronization of ovulation and copulation timings at AI, they succeeded in obtaining three-fold more fertilized embryos than with the previous method (based on natural mating with fertile males). Therefore, AI coupled with IASe may be useful for obtaining a large number of genetically engineered mice simultaneously.

## 4. SMGT-AI-Mediated Genome Editing (SMGT-AI-GE)

As mentioned previously, SMGT-AI is a highly convenient method for producing Tg animals. To the best of our knowledge, Sperandio et al. [91] were the first to demonstrate its usefulness in domestic animals, such as cattle and swine. When boar sperm cells obtained after pre-incubation with plasmid DNA were subjected to AI in ten sows, eighty-two offspring were obtained. Of these 82 offspring, 5 animals were identified as Tg, although the exogenous plasmid DNA included in their genomes was rearranged. In this study, it is suggested that SMGT-AI can be successfully adapted to generate Tg livestock. Yonezawa et al. [92] performed AI using rat epididymal sperm that had been incubated in a solution containing GFP-expressing plasmid DNA complexed with a liposome/human protamine-derived peptide. When morulae were isolated from the treated animals, GFP expression was observed. Furthermore, AI-treated animals produced pups carrying foreign DNA. More recently, Yina et al. [93] demonstrated that when the AI of mouse epididymal sperm that had been transfected with a linearized pRC/RSV vector carrying human clotting factor VIII cDNA with B-domain deleted (*BDD-hFVIII* cDNA) was applied to female mice, a total of nine F0 mice were delivered, in which three mice were identified as Tg. Moreover, the expression of *BDD-hFVIII* cDNA was detected in the livers and kidneys of all Tg offspring. These results suggest that SMGT is a rapid and convenient method for acquiring a large number of genetically engineered embryos through one-shot IVF.

Based on this, we considered that gene-engineered mouse epididymal sperm obtained after incubation with gene delivery-enhancing reagents for a short period (SMGT) could be applied to our novel surgical AI system (ITS) in mice, which is a simple method based on the intrabursal injection of epididymal sperm into superovulated females [69,70]. As depicted in Figure 4, isolated mouse epididymal sperm were first incubated in a solution containing the fluorescent marker expression plasmid DNA, RNP (containing Cas9 protein and gRNA), and gene delivery-enhancing reagents (such as DMSO, MagNPs, and ZIF-8) for a short period (SMGT). Then, a small volume (2 μL) of sperm (containing 2 × 10^5^ spermatozoa) were intrabursally injected 7 h after 5 IU hCG administration to females that had been administrated with a low dose of PMSG (<2 IU) 48 h before. During this process, exogenous DNA is transmitted to the oocytes via fertilization, leading to the creation of Tg embryos. To establish the expression of the transgene (plasmid) and occurrence of possible mutations in a target locus in earlier stages of experiments, the collection of two-cell embryos from AI-treated females one day after surgical AI may be preferable. The embryos were cultured until the early blastocyst stage. The expression of the transgene in blastocysts may be possible if the transgene contains a fluorescent gene, such as *EGFP*. Successful genome editing was also possible by analyzing genomic DNA isolated from blastocysts using a method previously described by Sakurai et al. [94]. Additionally, the presence of GE sequences in naturally delivered pups can be analyzed. Hereafter, we refer to this system as SMGT-AI-GE. Notably, this SMGT-AI-GE can also be performed using a non-surgical AI approach as recommended by Stone et al. [66], although a large volume (~40 μL) of sperm solution is required.

Recently, a report that the production of GE animals can be established using the SMGT-ET approach was first provided by Moradbeigi et al. [63], who used MBCD for accelerating the uptake of CRISPR/Cas9 reagents into sperm cells to generate targeted mutant blastocysts and mice. B6D2F1 (a hybrid between C57BL/6 and DBA/2) mouse sperm, were first incubated in the c-TYH medium (one of the IVF medium; a protein-free version of TYH medium established by replacing bovine serum albumin with polyvinyl alcohol [95] with 2 mM MBCD in the presence of 20 ng/µL pCAG-eCas9-GFP-U6-gRNA (pgRNA-Cas9) for 30 min). pgRNA-Cas9 is an all-in-one vector that contains an “enhanced specificity” *Streptococcus pyogenes*-derived Cas9 (eSpCas9) variant (which reduces off-target effects and maintains robust on-target cleavage) and 2A-GFP to the C-terminal of eSpCas9, together with a gRNA expression unit. The sperm suspension was then subjected to IVF in a drop containing oocyte-cumulus complexes. Moradbeigi et al. [63] showed that cholesterol removal from the sperm membrane using MBCD resulted in a premature acrosomal reaction, increased extracellular reactive oxygen species (ROS) levels, and a dose-dependent influence on the copy number of internalized plasmids per sperm cell. When successfully fertilized oocytes were allowed to develop in vitro, the resulting blastocysts exhibited GFP expression with efficiencies of over 80%. Sanger sequencing of the PCR-amplified products revealed the presence of indels at the target locus, with an efficiency of 25% (1/4). Additionally, when blastocysts obtained by incubating sperm with 2 mM MBCD and 20 ng/µL pgRNA-Cas9 were subjected to ET, one F0 mouse was obtained. Molecular biological analysis of the genome extracted from the kidney, liver, brain, and muscle revealed that the kidney and muscle had the same indel, but the brain had another indel, suggesting that the F0 mouse was a mosaic. This mosaicism may be ascribed to the use of an all-in-one CRISPR/Cas9 vector, because it takes 0.8–1.3 h for translation in embryos after fertilization with sperm carrying plasmids, and this time-lag may be the reason for mosaic embryo generation [96]. Moradbeigi et al. [64] suggested the use of an RNP system as an alternative to the plasmid system to avoid possible mosaicism, as RNP provides rapid and efficient action in DSB and indel creation [97]. Moradbeigi et al. [63] named this approach, methyl β-cyclodextrin-sperm-mediated gene transfer (MBCD-SMGT).

## 5. Conclusions

Here, we discuss the possible use of SMGT-AI-GE for the simple and convenient production of a large number of GE animals in a short period, as more than 10 ovulated females can be simultaneously subjected to AI using a small amount of sperm. This technology depends on the effective incorporation of genome-editing components into the sperm with the aid of gene delivery-enhancing reagents in vitro (SMGT) and in vivo fertilization of oocytes with transfected sperm (AI). SMGT-AI-GE will be especially valuable when researchers want to produce GE in large animals, such as pigs and cows, because surgical procedures are often difficult to perform in these individuals. On the other hand, AI is routinely used to propagate offspring in large animals. As previously shown by Sperandio et al. [91] succeeded in obtaining Tg domestic animals, such as cows and swine using SMGT and subsequent AI. This study encourages SMGT-AI-GE in future generations of GE livestock.

## Figures and Tables

**Figure 1 biotech-13-00045-f001:**
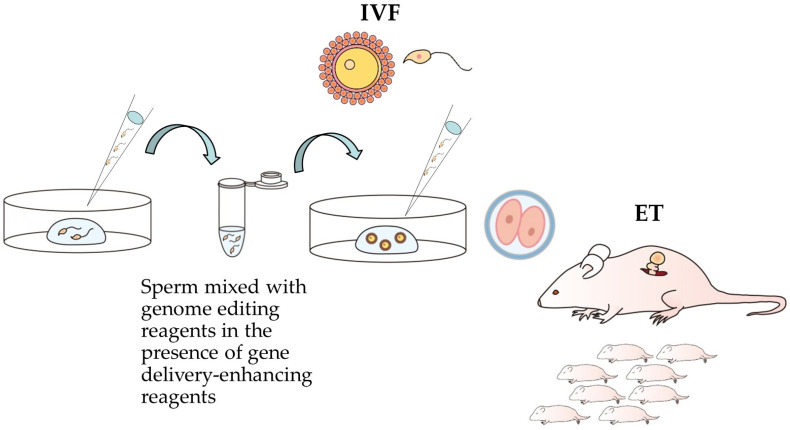
Schematic representation of sperm-mediated gene transfer–egg transfer (SMGT-ET). To obtain sperms transfected with exogenous nucleic acids, epididymal sperms were first isolated in a drop (in vitro fertilization (IVF) medium) covered with paraffin oil. Part of these sperms was then subjected to brief incubation with nucleic acids (i.e., plasmid DNA) and gene delivery-enhancing reagents (such as DMSO, liposomes, and nanoparticles). These transfected sperms were then subjected to IVF to obtain fertilized eggs, which were then cultivated up to two-cell embryos. These two-cell embryos are then transferred to the oviduct of a pseudopregnant recipient female for further development, ET.

**Figure 3 biotech-13-00045-f003:**
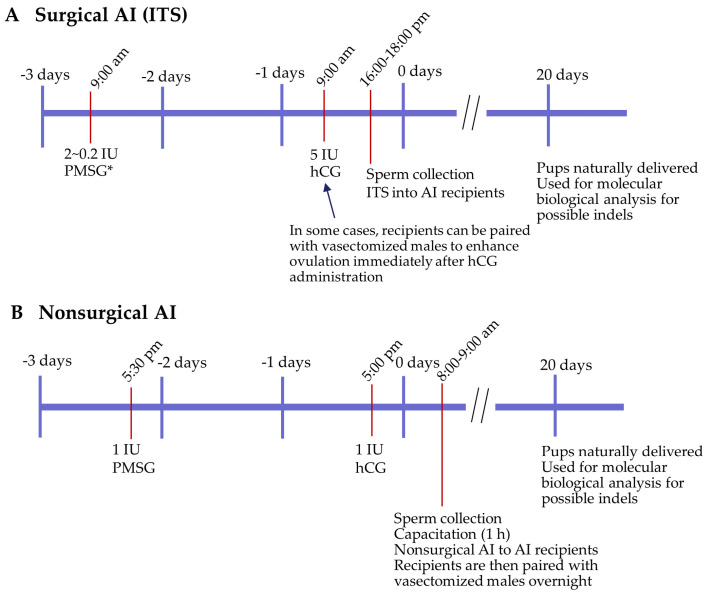
Time schedule for intrabursal transfer of sperm (ITS) (**A**) and non-surgical AI (**B**). The surgical AI (ITS) was conducted as previously established [71]. A low concentration of PMSG (2–0.2 IU) was administered at 9 AM to induce ovulation, similar to natural ovulation conditions (indicated by asterisks). The time schedule for non-surgical AI was based on Stone et al. [66].

**Figure 4 biotech-13-00045-f004:**
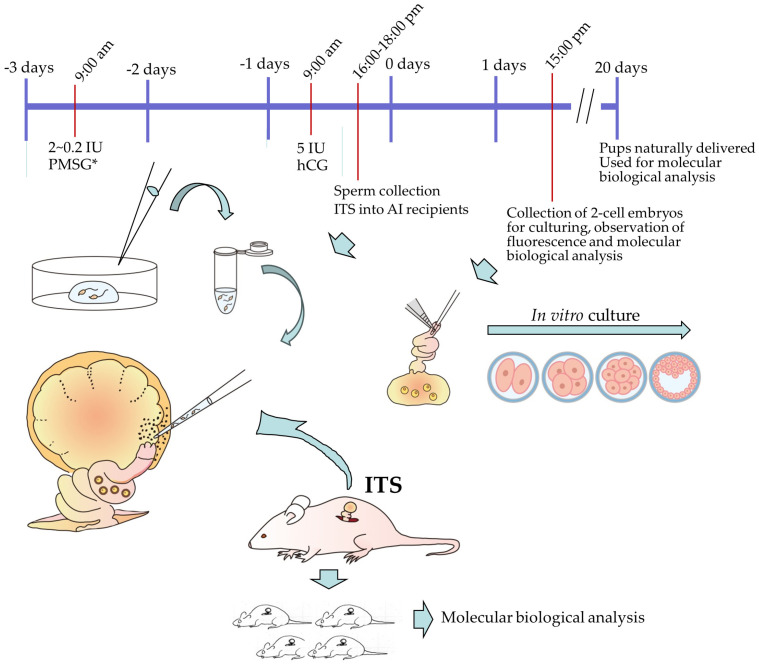
Schematic of sperm-mediated gene transfer–artificial insemination (SMGT-AI)-based genome editing (SMGT-AI-GE). To obtain sperms transfected with exogenous nucleic acids, isolated epididymal sperms were first subjected to brief incubation in a solution containing CRISPR/Cas9 components, gene delivery-enhancing reagents (such as DMSO, liposomes, and microparticles), and fluorescent marker expression plasmid DNA for a short period. The solution containing the transfected sperm was then subjected to AI (ITS) in superovulated females, 7 h after hCG administration. A low concentration of PMSG (2–0.2 IU) was administered at 9 AM to induce ovulation, similar to natural ovulation conditions (indicated by asterisks). Cleavage-stage embryos were collected to examine the presence/expression of the transgene (plasmid) and possible mutations in the target locus. In such cases, AI-treated females are permitted to deliver their pups. Genotyping of these pups may reveal genome editing at the target locus in F0 pups.

**Table 1 biotech-13-00045-t001:** Summary of the modified methods of sperm-mediated gene transfer (SMGT) for production of genetically modified animals.

Methods Used for SMGT or Reagents	Exogenous Nucleic Acids	Animals	Outcome	References
Linker-based SMGT (LB-SMGT)	pSEAP-2	Mice, pigs	Monoclonal antibody C (mAb C) is capable of binding to the surface antigen of sperm from all species, and exogenous DNA was used as a linker protein between the sperm and exogenous DNA. When this system was applied to porcine and mouse eggs, Tg offspring were obtained at efficiencies of 38% and 33%, respectively.	Chang et al. (2002) [56]
DMSO-SMGT	pEGFP-N1	Mice, rabbits	Mouse sperm were incubated in a solution containing 3% DMSO and 20 ng/µL of plasmid DNA for 10–15 min at 4 °C prior to IVF. The resulting embryos (42%) showed bright EGFP.	Shen et al. (2006) [58]
Retroviral vector-mediated SMGT	PLNCX2 carrying hLF DNA	Yak	The complex of PLNCX2-hLF + FuGene 6 complex was used to generate transduced yak spermatozoa. Oocytes inseminated with these sperm successfully developed to the blastocyst stage, suggesting the possible generation of Tg yaks.	Zi et al. (2009) [57]
MagNPs	pCX-EGFP/Neo	Boar	When boar sperm were incubated in the presence of 0.5% (*v*/*v*) MagNPs and plasmid DNA in a magnetic field for 90 min, and the magnetofected sperm were subjected to IVF with normal oocytes, the resulting fertilized eggs expressed EGFP.	Kim et al. (2010) [59]
REMI-SMGT	pEGFP	Rabbits	The AI of sperm incubated with *Bam* HI-digested plasmid DNA which had been complexed with liposomes or DMSO resulted in the generation of 14 newborn babies, of which 3 (one by restriction enzyme—liposome treatment and two by restriction enzyme—DMSO treatment) were found to be Tg.	Al-Shuhaib et al. (2013) [61]
MagNPs	pEGFP	Mice	Exogenous plasmid DNA loaded onto Fe_3_O_4_ magnetic nanocarriers (MagNPs) were delivered to mouse sperm cells under a magnetic field. IVF with these transfected sperms successfully led to the generation of Tg mice.	Wang et al. (2017) [60]
EFM DMSO	pEGFP-N1	Mice	When mouse sperm were incubated in electrolyte-free medium (EFM) or human tubular fluid (HTF) in the presence of DNA/DMSO, EFM was more effective for SMGT than HTF.	Kurd et al. (2018) [64]
ZIF-8	pCAG-eCAS9-GFP-U6-gRNA(pgRNA-Cas9)	Mice	When a mixture of ZIF-8 (60 ng/µL) and GFP-expressing plasmid (20 ng/µL) made by 15 min incubation was exposed to sperm for an additional 30 min incubation, and then IVF was carried out, the resulting blastocysts exhibited *EGFP*-derived fluorescence.	Sameni et al. (2024) [62]
MBCD	pgRNA-Cas9	Mice	When mouse sperm were incubated in IVF medium with 2 mM MBCD in the presence of 20 ng/µL plasmid DNA for 30 min and then the sperm suspension was subjected to IVF, the resulting blastocysts exhibited GFP expression with efficiencies of over 80%.	Moradbeigi et al. (2024) [63]

## Data Availability

Not applicable.

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
