# Peer review of "Artificial Insemination as a Possible Convenient Tool to Acquire Genome-Edited Mice via In Vivo Fertilization with Engineered Sperm"

_biotech, 2024, doi:10.3390/biotech13040045_

Round 1
Reviewer 1 Report
Comments and Suggestions for Authors
line 54
These indels often cause premature termination (stop) of codons ………..
Please make the sentence precise, for example:
These indels often generate stop codons causing premature termination …………...
line 54 ….[22]. The dot is not necessary.
Line 84 ………skilled technique……… change with “skilled stuff”
Line 109 …….bovine…………. Put noun instead of adjective
Line 127 …… (ET) ……. Change with (ET, embryo transfer)
Line 167 …… estrus or superovulated females………. Change with “using estrus or superovulation”
Line 324 ………….. 82 offspring were obtained. Of these five animals, the Tg was identified…………
The sentence is not clear. Maybe you mean “I five of these animals, the Tg was identified…………” ???
Comments on the Quality of English Languageline 54
These indels often cause premature termination (stop) of codons ………..
Please make the sentence precise, for example:
These indels often generate stop codons causing premature termination …………...
line 54 ….[22]. The dot is not necessary.
Line 84 ………skilled technique……… change with “skilled stuff”
Line 109 …….bovine…………. Put noun instead of adjective
Line 127 …… (ET) ……. Change with (ET, embryo transfer)
Line 167 …… estrus or superovulated females………. Change with “using estrus or superovulation”
Line 324 ………….. 82 offspring were obtained. Of these five animals, the Tg was identified…………
The sentence is not clear. Maybe you mean “I five of these animals, the Tg was identified…………” ???
Author Response
Reviewer-1:
Comments and Suggestions for Authors
Question-1: line 54: These indels often cause the premature termination (stop) of codons..
Please make the sentence more precise; for example, these indels often generate stop codons that cause premature termination.
Answer-1: As suggested, this portion has been corrected in the revised text (please see L54-55).
Question-2: line 74.[22]. Dots were not necessary.
Answer-2: This portion has been corrected in the revised text (please see L100).
Question-3: Line 82 ………skilled technique……… change with “skilled stuff”
Answer-3: This portion has been corrected in the revised text (please see L102).
Question-4: Line 107 …….bovine…………. Placing nouns instead of adjectives
Answer-4: This portion has been corrected in the revised text (please see L133).
Question-5: Line 125 …… (ET) ……. Changes in (transfer and embryo transfer)
Answer-5: The term ET has already been used in L107 as embryo transfer (ET). Therefore, this portion was expressed as “--- which is called “ET, “a time-consuming ---” in the revised text (please see L221).
Question-6: Line 167 … estrus or superovulated females…. Change with “using estrus or superovulation”
Answer-6: This portion was changed to --- who used females in estrus or superovulated female mice to develop a simple user-friendly AI. Please see L261-263 of the revised manuscript.
Question-7: Line 321 ………….. 82 offspring were obtained. Of these five animals, the Tg was identified.
However, this sentence was unclear. Maybe you mean “I five of these animals, the Tg was identified…………” ???
Answer-7: We thank the reviewer for highlighting this oversight. We have corrected this portion as follows: --- 82 offspring were obtained. Of these 82 offspring, five animals were identified as Tg (please see L362-363 in the revised text).

Reviewer 2 Report
Comments and Suggestions for Authors
This work is very well developed and documented. Advances in genome editing technology have made it possible to create animals with edited genomes to produce models for studying human diseases. In this review, artificial insemination has been compared with in vitro fertilization, the latter being more commonly used for this type of study. A new method has been proposed using artificial insemination, which is much simpler and more cost-effective. The proposed results are promising and open up new and very interesting avenues for research.
Perhaps the only drawback would be to use a more up-to-date bibliography, especially regarding genome editing techniques, such as for example.
Qian Y, Wang D, Niu W, Shi Z, Wu M, Zhao D, Li J, Gao X, Zhang Z, Lai L, Li Z. Development of a highly efficient prime editor system in mice and rabbits. Cell Mol Life Sci. 2023 Nov 4;80(11):346.
Author Response
Reviewer-2: Comments and Suggestions for Authors
This work has been well developed and documented. Advances in genome editing technology have made it possible to create animals with edited genomes as models for studying human diseases. In this review, artificial insemination has been compared with in vitro fertilization, the latter being more commonly used for this type of study. A new method using artificial insemination has been proposed that is simpler and more cost-effective. The proposed results are promising and open new interesting avenues of research.
Question-1: Perhaps the only drawback would be to use a more up-to-date bibliography, especially regarding genome editing techniques, such as.
Qian Y, Wang D, Niu W, Shi Z, Wu M, Zhao D, Li J, Gao X, Zhang Z, Lai L, Li Z. Development of a highly efficient prime editor system in mice and rabbits. Cell Mol Life Sci. 2023 Nov 4;80(11):346.
Answer-1: We thank the reviewer for identifying the discrepancy in our citation. We have now updated our citations to reflect the most recent research. In the revised text, we have added information on new tools (BE and PE) that enable precise genome editing of a target gene (please see L69-93).

Reviewer 3 Report
Comments and Suggestions for Authors
This review is quite interesting though I have a few recommendations.
1. The title should say mice as opposed to mammals as 95% of the data is very specific for these animals.
2. The authors should consider placing section 3 after section 1 as section 2 deals with later events (the subsequent use of sperm, the modification of which is described in section 2).
3. The review is all about genome editing yet section 3 gives very little details about genome modification, while a huge section is dedicated (section 2) to getting sperm back into specifically mice. Controversies are alluded to but not discussed in any detail. This part needs to be the main focus of the entire review. What is the point of being able to fertilise mice in vivo if the sperm have not actually been modified? Thus the data that this works (or not) needs to be exhaustively analysed.
4. Section 3 needs a table that discusses the pros and cons of the various sperm mediated GM methods with references.
Minor comments:
l70: towards?
l86 ..editing in early mouse embryos viaoviductal
l115 missing the references
l142 and later mouth-controlled
l140-142 rewrite sentence starting with "Using a"
l217 41 vs 21 per mouse
l236 when trial 14 makes no sense
l279 delete among
l291 write out EP
l312..embryos, required transfer to recipient females,
l325 The transgene was identified in five of these animals...
l335 a total of 9FO etc does not make sense
l375 write out MBCD
l403-5 unclear what the authors mean to say
l408 Here we discuss
Author Response
Reviewer-3: Comments and Suggestions for Authors
Although this review is interesting, there are a few recommendations.
Question-1: The title should say that mice, as opposed to mammals, as 95% of the data are very specific for these animals.
Answer-1: This has been corrected in the revised manuscript (please see L3).
Question-2: The authors should consider placing section 3 after section 1, because section 2 deals with later events (the subsequent use of sperm, the modification of which is described in section 2).
Answer-2: We greatly appreciate the feedback. As suggested, the order of the sections has been changed in the revised manuscript.
Question-3: The review is all about genome editing, yet section 3 provides very little detail about genome modification, while a huge section is dedicated (section 2) to getting sperm back into specific mice. Controversies have been alluded to but have not been discussed in detail. This section is the focus of the entire review. What is the ability to fertilize mice in vivo if the sperm has not actually been modified? Thus, the data used in this study (or not) needs to be exhaustively analyzed.
Answer-3: We appreciate the reviewer’s comments about the focus of the review. As suggested, Section 3 was placed after Section 1 and is now shown in Section 2. In this section, a more detailed description of the modified SMGT is provided in the revised text (see L174-192).
Question-4. Section 3 presents a table that discusses the advantages and disadvantages of various sperm-mediated GM methods with references.
Answer-4: As suggested, Table 1 was developed, in which the modified methods of SMGT are listed chronologically.
Minor comments:
L70: towards?
Answer: This portion was deleted in the revised text (see L95).
L86 ..editing in early mouse embryos via oviductal
Answer: This portion was corrected in the revised text (see L111).
L115 missing the references
Answer: This portion was corrected in the revised text (see L39-140).
L142 and later mouth-controlled
Answer: This portion was corrected in the revised text (see L235).
L140-142 rewrite sentence starting with "Using a"
Answer: This portion was corrected in the revised text (see L235-237).
L217 41 vs 21 per mouse
Answer: This portion was corrected in the revised text (see L312-313).
L236 when trial 14 makes no sense
Answer: This portion was deleted in the revised text (see L329-331).
L279 delete among
Answer: This portion was deleted in the revised text (see L149).
L291 write out EP
Answer: This portion was deleted in the revised text (see L170-173).
L312..embryos, required transfer to recipient females,
Answer: This portion was corrected in the revised text (see L205-207).
L325 The transgene was identified in five of these animals...
Answer: This portion was corrected in the revised text (see L362-363).
L335 a total of 9 FO etc does not make sense
Answer: This portion was corrected in the revised text (see L373-374).
L375 write out MBCD
Answer: This abbreviation is already shown in the revised text (see L173).
L403-5 unclear what the authors mean to say
Answer: This portion was deleted in the revised text (see L440).
L408 Here we discuss
Answer: This portion was corrected in the revised text (see L443).
